# E-Learning Performance Evaluation in Medical Education—A Bibliometric and Visualization Analysis

**DOI:** 10.3390/healthcare11020232

**Published:** 2023-01-12

**Authors:** Deborah Oluwadele, Yashik Singh, Timothy T. Adeliyi

**Affiliations:** 1Department of Telemedicine, School of Nursing and Public Health, University of KwaZulu-Natal, Durban 4041, South Africa; 2ICT and Society Research Group, Information Technology, Durban University of Technology, Durban 4001, South Africa

**Keywords:** bibliometric analysis, e-learning, medical education, performance evaluation

## Abstract

Performance evaluation is one of the most critical components in assuring the comprehensive development of e-learning in medical education (e-LMED). Although several studies evaluate performance in e-LMED, no study presently maps the rising scientific knowledge and evolutionary patterns that establish a solid background to investigate and quantify the efficacy of the evaluation of performance in e-LMED. Therefore, this study aims to quantify scientific productivity, identify the key terms and analyze the extent of research collaboration in this domain. We searched the SCOPUS database using search terms informed by the PICOS model, and a total of 315 studies published between 1991 and 2022 were retrieved. Performance analysis, science mapping, network analysis, and visualization were performed using R Bibliometrix, Biblioshiny, and VOSviewer packages. Findings reveal that authors are actively publishing and collaborating in this domain, which experienced a sporadic publication increase in 2021. Most of the top publications, collaborations, countries, institutions, and journals are produced in first-world countries. In addition, studies evaluating performance in e-LMED evaluated constructs such as efficacy, knowledge gain, student perception, confidence level, acceptability, feasibility, usability, and willingness to recommend e-learning, mainly using pre-tests and post-tests experimental design methods. This study can help researchers understand the existing landscape of performance evaluation in e-LMED and could be used as a background to investigate and quantify the efficacy of the evaluation of e-LMED.

## 1. Introduction

The need to develop the body of knowledge in performance evaluation of e-learning in medical education is quite daunting to explain. E-learning has ceased to be an option in higher education institutions due to the advent and spread of COVID-19. During the hard lockdown, physical and social distancing was adopted as measures to decrease the transmission of the virus in dense populations such as higher education institutions [1]. This disrupted the teaching-learning processes in more than 90% of the world’s population of students in the education system [2]. As a result, many universities took desperate measures and suddenly transitioned from traditional face-to-face learning to e-learning [3], a process termed Emergency Remote Learning (ERL) rather than e-learning. This solution is stated by the literature as an “imperfect yet quick solution to the crises” [2] and could impair student performance. 

It is recommended to follow a systemic approach when adopting e-learning in Medical Education, the first step being the assessment of needs and thorough requirement engineering [4]. However, the readiness index for e-learning implementation in many universities is low. This is evident in the lack of human resources, technology infrastructure, learning management systems, and student support structures. The lack of preparation in the implementation of e-learning causes several challenges [5] and calls for cross-disciplinary research into the development of evaluation models for e-learning [6], especially in the medical education domain.

Concomitantly, the effect of COVID-19 cannot be discussed without mentioning the ripple effect of the pandemic in the medical domain. The pandemic necessitated an increased demand for medical practitioners to manage the burden of diseases and, consequently, a growing need to strengthen the capacity of healthcare professionals. Although e-learning is stated to be effective in enhancing the capacity of healthcare workers, the lack of a systematic approach to the design, monitoring, and evaluation of e-learning makes its impact on medical education highly debatable. 

Very little research has been conducted to determine the key factors that impact student performance or to design a model for student performance evaluation in e-LMED. However, in the broader e-learning context, a handful of studies have been conducted on evaluating e-learning [6]. These studies used different techniques, methods, and approaches for assessing students’ performance [7,8,9,10,11]; therefore, the ubiquity of publications has resulted in the further divergence of the body of knowledge in this domain. This fact is reinforced in the systematic review conducted by De Leeuw, De Soet [12]. The study reported that the evaluation of performance in e-learning is very complex due to the assortment of e-learning methods and the diverse approaches to carry out such evaluation correctly. The study further articulated that the domain is yet to achieve any form of consensus about which indicators to evaluate and calls for further studies to develop an evaluation tool that is properly constructed, validated, and tested. This is perceived as a firm footing for researchers to compare their findings on e-learning performance evaluation and for continuous improvement of the body of knowledge in the domain.

This study explores the grey area identified above by pooling together the published studies, aiming to create a convergence of the body of knowledge in the performance evaluation of e-learning in medical education. Hence, the following questions are posed: What authors, countries, institutions, and journals have actively published on performance evaluation in e-LMED? What are the most recurring keywords in the published literature on performance evaluation in e-LMED? Are authors from different disciplinary backgrounds working together to publish on performance evaluation in e-LMED?

The Population, Intervention, Comparison, Results, and Study Design (PICOS) model were used to identify publications relevant to this study’s aim on the SCOPUS database. Scopus was used because it is 100% inclusive of MEDLINE and has a more significant number of indexed journals than other databases. Also, SCOPUS has many functions that can be leveraged to facilitate citation analysis, counting research collaboration, and exporting data to Microsoft Excel for further tabulation and mapping. The bibliometric analysis method was used to analyze the retrieved documents.

This paper presents a brief contextual background to our study by exploring the definition of e-learning and the methods, models, and theories used to evaluate performance in e-learning in medical education. The next part of the paper presents our method and study design and then the results of our analysis. We conclude the paper with the discussion and conclusion sessions where we interpret the results and highlight the limitation and make recommendations for future research. 

### Short Contextual Background of the Study

The discussion about the benefits of e-learning in medical education has been ongoing before the advent of the COVID-19 pandemic. From different perspectives, proponents of e-learning have stated the potential benefits of e-LMED. Sears et al. [13] opined that individuals involved in e-learning had a better ability to apply knowledge and skills and to retain learned concepts in a professional setting over a long period. E-learning allows medical students to study across borders at remote locations at their convenience while giving them access to a vast array of academic resources [14]. The core benefit of e-learning is that it facilitates learning without taking Healthcare Professionals (HCPs) from their locations or working environment, as this could further burden the already burdened system [15].

Web 2.0 induced a paradigm shift in e-learning ten years ago and enabled e-learning scenarios that were precursors to the current e-learning landscape. This is characterized by dynamic technological environments that allow users to create their own content and collaborate with other users [16]. Thurzo, Stanko [17] evaluated the effect of web 2.0 on dental education. The findings from the study revealed an increasing number of e-learning resources based on WEB 2.0 innovative technologies. Presently, the socialization of the internet with the advent of Web 4.0 and the anticipation of Web 5.0 with artificial intelligence capabilities [18] will redefine the prospects of medical education. With these claims, e-learning presents as a powerful and timely pedagogical tool in the current COVID-19 context, introducing further resource and geographical constraints. Even though there are several works of literature investigating e-learning adoption and usage in medical education, these studies evaluate different performance constructs using various e-learning tools within different disciplinary contexts. No study brings all these variations into one literature so that the body of knowledge in this domain starts to grow. 

E-learning is learning instructions delivered on digital devices such as desktop computers, laptops, tablets, or smartphones to support learning [19]. E-learning is utilized in the modern-day teaching and learning process to support education, improve knowledge, advance performance, and improve students’ learning outcomes. For this study, we operationalize performance as the degree of efficiency and effectiveness with which a student carries out his assigned tasks. Efficiency in this context refers to obtaining results with limited resources; effectiveness is the achievement of the desired goals [20]. 

On the surface, the evaluation of the performance of e-learning intervention seems to be concerned with just the technology, or at most, the technology, together with the task requirements for which the technology was adopted. However, other factors may influence student performance on an individual and organizational level. Several models have been developed which describe the effect of technology on performance, yet, the contradiction in the realization of the expected benefits of technology calls for a deeper understanding of this effect [20]. Most research that evaluates performance focus mainly on a single component (technology, task, or individual); nevertheless, these studies do not provide factors that need to be considered, monitored, and evaluated for assessing students’ individual and organizational performance in medical education.

Also, the previous studies in the domain are conducted in different disciplines or departments in medical education; the e-learning tools and platforms used to facilitate learning are diverse, and so are the various outcomes reported. Ajenifuja and Adeliyi [21] and Oluwadele [22] assessed the influence of e-learning on the performance of healthcare professionals pre-covid. The study designed a hybrid framework by combining the Task-Technology fit model [23] and the Kirkpatrick evaluation model [24]. It postulates that when there is alignment between the learning tasks, technology infrastructures, individual characteristics, and contextual characteristics of students in medical education, their performance in e-learning will be optimized. Performance in this study was operationalized using the four constructs of the Kirkpatrick evaluation model—reaction, learning, behavior, and result. The study evaluated the performance of students who participated in an online antimicrobial stewardship and conservancy module hosted by the University of Kwazulu-Natal, South Africa, from four different perspectives according to the four constructs of Kirkpatrick evaluation model. Reaction (satisfaction with the module), Learning (scores in the module), Behavior (ability to apply learned concepts at work), and Result (the impact of knowledge transfer to practice in the workplace).

The result found that performance was enhanced both at the individual (result and learning) and organizational level (behavior and result) because the technological infrastructure provided to facilitate the module aligned with the task requirements of the module. Furthermore, participants affirmed that the lecturers and support team provided support to mitigate negative individual and contextual characteristics which could have hindered their performance. This includes a translator to moderate the language barrier non-English speaking students encounter and training for participants without experience in e-learning. Interestingly though, participants echoed that they did learn not only the content of the e-module but also acquired technical and research capabilities, which they have found to be even more helpful in their daily work practice.

Other researchers in the e-learning evaluation domain operationalize performance in different ways and use different models to evaluate performance in e-learning. For instance, Tautz, Sprenger [25] operationalized e-learning performance with repetition and active learning in university classrooms and used the DeLone and McLean model to measure constructs such as quality of system use, perceived benefits, and student perspectives. Prasetyo, Ong [26] operationalized e-learning performance as the acceptance of e-learning platforms. The study evaluates e-learning performance using the Extended Technology Acceptance Model (ETAM) and DeLone and McLean Information Systems Success Model. Constructs such as user interface, perceived ease of use, perceived usefulness, information quality, system quality, behavioral intentions, and actual usage of the system were measured. 

Mastan, Sensuse [6] reported on the models used by researchers for e-learning evaluation. This includes the 5 Dimension Evaluation Model [27], the Kirkpatrick Model [28], the System Usability Model [29,30], the Technology Acceptance Model (TAM) [30,31,32], Swot Analysis [31], the Balanced Scorecard (BSC) [33], Theory of Planned Behavior (TPB) [31], Expectation Confirmation Model (ECM) [31], Flow Theory [31], and E-learning System Success Model [34]. While these models have been used to evaluate e-learning platforms, systems, tools, and interventions, student performance or outcome evaluation is not the focus of most of these studies.

Studies evaluating e-learning performance in medical education are diverse, and there seems to be no point of convergence on the constructs to be assessed so that performance is understood. Besides, medical education is a complex domain that does not yield itself easily to the adoption and utilization of e-learning as much as other domains. Therefore, this study fills the gap by accessing the published literature on performance evaluation in e-LMED to quantify the literature, identify the key terms and understand the extent of research collaboration in performance evaluation in e-learning in medical education.

## 2. Materials and Methods

This study aims to quantify scientific productivity, identify the key terms and analyze the extent of research collaboration in this domain by conducting a bibliometric analysis of publications on performance evaluation in medical education. 

### 2.1. Database

Scopus database was used to retrieve, analyze and map data, and provide information about citation and research collaboration related to performance in medical education. Data for the bibliometric analysis were extracted using the PICOS model. We identified and retrieved 315 relevant document documents. 

### 2.2. Search Strategy

The PICOS (Population, Intervention, Comparisons, Outcomes, and Setting) model ensures scientific diligence and objectivity of reviews by prescribing methodological standards that enhance the value of the scientifically published literature reviews and guarantee their robust reproducibility [35]. Using the PICOS model, we highlighted the population or participants as e-learning—any online course from across the world. Our focus on e-learning was defined by the intervention or exposure construct of the PICOS model. Here, we looked for the literature focusing on evaluating performance in an e-learning context. The Comparison and Outcome constructs were not applicable in our context because of the aim of our study. However, the Setting construct defined the context within which we would be considering the intervention—Medical education. Consequently, the PICOS model streamlined our search by informing the progression of our thought process from the conceptual phase to the logical and physical phases, where we developed the criteria for studies to be included in the analysis and, subsequently, the main keywords to be used for creating the search terms. Table 1 presents the inclusion criteria for the review using the PICOS model.

An initial search was conducted on Scopus on 16 May 2022 using keywords and relevant synonyms from the general population, then the intervention and setting. These keywords were obtained by reading published works and noting key terms related to e-learning in the medical education domain. The search was modified as many times as possible to sharpen the result, increase the validity of the search strategy and ensure minimum false-positive and false-negative results. 

Table 2 shows the keywords used from the initial to the final searches and the results returned.

### 2.3. Validation and Quality Assurance of the Search Query

The synonyms of the key search terms were researched and included to ensure maximum inclusivity of published work in the domain. After this, the keywords were modified in several iterations to confirm the validity of the search strategy. The modification helped to eliminate false-positive and false-negative results. The first document results were analyzed to ensure they aligned with the scope of the study, thereby reducing the possibility of false-positive results. 

For false-negative results, the number of documents for the top active authors shown in the Scopus database was compared with their research profile in Scopus to assess the extent of agreement between what has been retrieved and what is actually in the Scopus database about the desired research question. 

### 2.4. Data Analysis and Visualization

Citation analysis, co-word analysis, and co-author were used to analyze the retrieved data. Citation analysis of a research field examines the most cited studies, authors, or journals, typically in the form of top-N lists. It is perceived as a measure of influence if an article is popularly cited. This assumption is based on the perception that authors cite documents they consider essential for their work. While Citation analysis provides information about the relative influence of the publications, it cannot recognize networks of interconnections among scholars. Thus, Co-author analysis was used to give an indication of collaboration and produce the social structure of the field. At the same time, Co-word analysis was used to find connections among concepts that co-occur in document titles, keywords, or abstracts [32]. 

The retrieved data were analyzed using Biblioshiny, a bibliometric package on R, while the visualization was performed using VOSviewer for bibliometric indicators such as annual growth, active authors and their collaboration, journals and countries, the frequently used keywords, fields and subject areas. The retrieved documents were analyzed for bibliometric indicators such as the growth of publication, active authors and their collaboration, countries, and institutions most actively involved in the publication, active journals, frequent author keywords, the geographic distribution of the literature, and the target fields and subject areas.

## 3. Results

### 3.1. Citation Analysis

Citation analysis was used to examine the most cited studies, authors, or journals, typically in the form of top-N lists. If an article is popularly cited, it is perceived as a measure of influence. This assumption is based on the perception that authors cite documents they consider essential for their work. The citation analysis of this study was conducted using the analysis functionality on Scopus.

#### 3.1.1. General Description of the Retrieved Publications

The search query returned 315 documents. The documents were not limited to any language or type to ensure maximum inclusivity of all works published in this domain. Figure 1 shows the characteristics of the retrieved documents. The retrieved documents consist of 257 Articles, 25 Conference Papers, 19 Reviews, 7 Letters, 3 Notes, 3 Short Surveys, and 1 Editorial.

The dominant disciplines (Figure 2) for the publication were medicine (n = 211), next was Social Sciences (n = 103), followed by Health Professions (n = 31), Engineering (n = 17), Nursing (n = 16), Computer science (n = 11), Biochemistry, Genetics and Molecular Biology (n = 11).

#### 3.1.2. Growth of Publication

The retrieved documents were published between 1991 and 2022. The first article in the performance evaluation domain was published by Medical Teacher in the social science subject area. The authors were Kamien, Macadam [36]. The next paper in this domain was only published in 1995 (n = 3) and 1996 (n = 6), after which nothing was published until 1999 (n = 1). However, publication in this domain continued to grow year after that and witnessed sporadic growth in 2021 (n = 47, 14.92%); The most popular document published in 2021 was the work conducted by Prigoff, Hunter [37]. Figure 3 shows the growth of publications in this domain.

#### 3.1.3. Top Publishing Author

In total, 1598 authors contributed to the retrieved documents, giving an average of 5.07 authors per document; 21 (1.3%) were authors of single-authored documents, while 1577 (98.7%) were authors of multi-authored documents. Figure 4 shows the top ten publishing authors. Four authors are the first, each producing four (1.27%) documents in this domain. These are Professor Arvanitis, T. N., a professor of Digital Health Innovation and Director of the Institute of Digital Healthcare, WMG from the University of Warwick, Coventry, United Kingdom Horvath, Rita from the University of Cambridge, United Kingdom, Khan K.S from the Universidad de Granada, Granada, Spain and Kunz, R. from Universitat Basel, Basel, Switzerland. These four authors published in this domain via BMC Medical Education between 2007 and 2009.

#### 3.1.4. Most Active Country

Figure 5 illustrates the top ten most actively publishing countries in performance evaluation of e-LMED. United States (U.S.) takes the lead with 119 (37.8%) publications, followed by the United Kingdom (UK) (n = 50; 15.87%) and Canada (n = 25; 7.9%). Germany (n = 24; 7.61) and Australia (n = 22; 6.98%) have over 20 publications, while the remaining countries have less than 20 publications. The U.S. has been publishing in this domain since 1995, the U.K. in 1991, Canada in 2005, Germany in 2006, and Australia in 1991.

#### 3.1.5. Most Active Institutions

We determine the most active institutions by the number of documents published by the institution. The University of Sydney (n = 7; 2.22%) is the most active institution, followed by Harvard Medical School, the University of Toronto, Memorial University of Newfoundland, and the University of Birmingham (n = 6; 1.90%)(Figure 6). The University of Sydney has been publishing since 2003, Harvard Medical School since 2012, the University of Toronto since 2006, the Memorial University of Newfoundland since 2005, and the University of Birmingham since 2006.

#### 3.1.6. Most Active Source

The retrieved documents were published in 186 different sources. BMC Medical Education; is a UK-based open-access journal active since 2001 and publishing original peer-reviewed research articles concerning the training of healthcare professionals, including undergraduate, postgraduate, and continuing education with a specific focus on curriculum development, evaluations of performance, assessment of training needs and evidence-based medicine ranked as the most active source (n = 30; 9.5%). This is followed by Medical Teacher (n = 19; 6.0%), also a UK-based journal active since 1979 and addressing the needs of teachers throughout the world involved in training for the health professions. 

Studies in Health Technology and Informatics—based in Netherland-based journal focused on Biomedical Engineering, Health Information Management, and Health Informatics ranked as the third most active source (n = 11; 3.5%), Anatomical Sciences Education; a US-based journal active since 2008 providing an international forum for the evidence-based exchange of ideas, opinions, innovations, and research on topics related to education in the anatomical sciences ranked fourth (n = 9; 2.9%) while Journal of General Internal Medicine ranked fifth (n = 7; 2.2%). Together, these top five most active sources contributed approximately 24% of the documents in performance evaluation in e-learning.

Figure 7 is a line graph of the top ten most active sources in performance evaluation of e-learning in medical education relative to the year and volume of publication. Even though Medical Teacher ranked as the second most active source, it was the first to publish in the domain. The article published by Kamien, Macadam [36] describes the development and evaluation of a structured introductory course in general practice. However, the higher yearly publication per source in the domain so far is seven documents published by BMC Medical Education in 2016.

### 3.2. Co-Word Analysis

Co-word analysis was used to discover the connections and interrelationships that exist among concepts that co-occur in document titles, keywords, or abstracts of the retrieved document [38]. Co-word analysis exposes central issues connected to the performance evaluation of e-LMED through the analysis of re-occurring keywords and topics in the domain. The connections between these keywords are expressed in terms of the number of occurrences and the Total Link Strength (TLS) of the keywords.

#### 3.2.1. Most Frequent Author Keywords

To gain insight into the conceptual structure of the domain, we analyzed the most frequently encountered author keywords. Seven hundred keywords were detected in the retrieved documents. To ensure meaningful visualization, we limit the keyword to 20 using a threshold of a minimum number of occurrences of a keyword to 5. For each of the 20 keywords, the TLS with other keywords was calculated, and the keyword with the greatest total link strength was selected. The keywords with the highest occurrence and TLS were e-learning (n = 69), medical education (n = 45), online learning (n = 25), distance learning (n = 25), and education (n = 26). Table 3 shows the top 20 keywords in terms of the frequency of occurrences and TLS, while Figure 8 is an overlay visualization map of the top 20 frequent author keywords representing approximately 1% of the total author keywords (n = 700) in the retrieved articles. The map shows the progression of keywords, with the keywords in purple being the older ones and the yellow being the most recent keywords in the literature. While the initial keywords in the domain include distance learning, internet, continuing medical education, web-based learning, and evaluation, the most recent keywords were COVID-19, residency, and flipped classroom.

#### 3.2.2. Term Co-Occurrence

The term co-occurrence map depicts the relationship and interconnection between different terms based on the paired presence of the terms. When two or more terms occur together, it means there is a relationship between those terms. Terms from the title and abstract fields of the retrieved document were extracted, and the full counting method was used. There were 8550 terms in total. However, we chose a threshold of a minimum number of 10 occurrences per term for our analysis, and only 244 (2.9%) terms met the threshold. A relevance score was calculated for each of the 244 terms, and 146 (60%) of the most relevant terms were selected by default based on this score. 

Figure 9 and Figure 10 present a Network and Overlay Visualization of the most relevant terms in the performance evaluation of e-LMED, while Table 4 shows the 20 most pertinent terms of the domain according to the frequency of their occurrence and the relevance of the terms. The network (Figure 9) consists of eight color-coded clusters with node sizes representing the frequency of occurrence of each term and links representing the relevance of and co-occurrence between different terms within a cluster. The terms with the more prominent nodes, such as e-learning, residents, systems, performance, covid, and resident, are terms with a higher frequency of occurrence in different clusters. 

Figure 9 and Figure 10 are the same, however, Figure 10 analyzes the evolution and dominance of these terms relative to time. The terms color-coded in dark purple were common in performance evaluation of e-LMED around the year 2010. These include distance learning, system, computer, CME, EBM, man, woman, and cost, to mention just a few. Terms color-coded in green were common around 2012–2014. These include e-learning, access, evidence, opportunity, factor, barrier, information, and so on. The latest terms in the domain are terms color-coded in yellow. These include residents, session, performance, intervention, pandemic, covid, person, platform, and flipped classroom.

Specifically, the term “performance” was highlighted to understand its co-occurrence network. This will help us conceptualize and draw inferences about terms that are related to performance in the medical education domain. Figure 11 shows the terms that co-occur with performance and the closeness of these terms. These include perception, student performance, pre-test, post-test, platform, flipped classroom, pandemic, covid, e-learning, factor, session, and intervention. These terms suggest the evolutionary trends of how performance is perceived, evaluated, or studied in e-learning for medical education and give a possible research direction that might be explored further in the future.

#### 3.2.3. Conceptual Structure Map-Method: MCA

Typically, concepts are embedded in a network of associations, and the meaning of a concept can be traced in part to the other concepts linked to them. Hence, we examine the conceptual structure of performance evaluation of the e-LMED domain by conducting a factorial analysis using the Multiple Correspondence Analysis (MCA) methods on Biblioshiny. Using K-means clustering: an unsupervised classification to identify clusters in the retrieved document that express common concepts with the relative location of the dots and their distribution along the dimensions, the findings categorize keywords in the domain into two clusters. The first cluster creates a network of keywords that are closely associated. Problem-based learning and organization and management are some of the most frequently encountered topics and are closely associated. The second cluster consists of keywords that may be considered as the different constructs and tools used in the performance evaluation of e-learning in medical education. This includes health personnel attitude, methodology, healthcare quality, and questionnaires (Figure 12).

#### 3.2.4. Topic Dendrogram

The dendrogram represents the hierarchical order and relationship between the keywords generated by hierarchical clustering. The dendrogram groups the keywords into two clusters using the content of the retrieved document. The underlying method is that when words frequently co-occur in documents, the concepts behind those words are closely related. Figure 13 shows a network of themes and their relations representing the conceptual space of performance evaluation of e-LMED. Cluster 1 classifies health care quality, methodology, questionnaires, the attitude of personnel, and health personnel attitude together, while Cluster 2 group previously identified keywords and terms in the domain such as e-learning, pandemic and COVID-19, course evaluation, continuing education to mention just a few. 

#### 3.2.5. Factorial Map of the Most Contributing Papers

Figure 14 provides analytical insight into the document with the highest contributing papers related to the clusters identified in the word map and the topic dendrogram. Cluster 1 shows six papers, while cluster 2 also shows six papers.

Cluster 1: The first cluster identified the work by Nagaraj, Yadurappa [39], the effectiveness of blended learning in radiological anatomy for first-year undergraduate medical students, as one of the papers with the highest contributions. The study assessed the efficacy of blended learning by estimating knowledge gain and evaluating student perceptions. A single group pre-and post-test was used to assess five anatomy modules consisting of online presentations and self-assessment quizzes uploaded to a Learning Management System (LMS). Students peruse the module via the LMS while the teacher facilitates the module physically in the classroom. The study found a significant difference between the pre-and post-test score and affirmed that the students positively perceived the module. Smith and Boscak [40] described their experience restructuring a four-week Trauma and Emergency Radiology Elective course for third- and fourth-year medical students to an online format. The major changes to the module include the assignment of loads of self-study educational resources, independent review of unknown cases using a virtual workstation, and online interactive conferences. Students completed post-course feedback surveys, and the result showed that students perceived the course materials as clinically relevant, accessible, and engaging. They learned from the module, as suggested by the increase in confidence in ordering and interpreting imaging studies, and they were willing to recommend this rotation to other students. However, 60% of the students said they would still prefer blended learning over e-learning. Phillips, Edwards [41] examined the acceptability, feasibility, and proof of concept of slacks- an online channel-based messaging app for undergraduate dermatology education. Undergraduate medical students participated in an online classroom for a six-week program consisting of case-based discussions, seminars, and journal clubs. Student doctors and patient educators facilitated the platform while students and faculty completed a post-course evaluation survey. The study recorded a low participation level with more than 50% of the participants indicating that they used the platform as passive observers. The study opined that a community-based online classes may serve as an enjoyable, acceptable and collaborative means of delivering dermatology education to undergraduate medical students. Matthews, Tian [42] conducted a usability study of an in-use emergency medicine V.R. training application using nine users with no prior V.R. experience but with relevant expertise. The study found an above-average usability score and significant improvement in several factors, including performance. Nilsson, Östergren [43] investigated if the individual learning style of medical students influences their choice to use a web-based ECG learning program in a blended learning setting. The study found that neither learning style nor other characteristics appeared to influence students’ choice of web-based ECG program and suggested that web-based learning may attract a wide variety of medical students. 

Cluster 2: The second cluster identified six papers dating back to the pre-covid era when distance education was used to denote e-learning. These studies explored the feasibility, implementation, and outcomes of e-learning in medical education across different countries and continents. Klein, Hannum [44] explored and reported on resource sharing through distance education to solve the lack of resources in orthodontic residency programs in the United States and Canada. A blended approach to distance learning was used. Despite the challenges experienced, the study ascertained that the approach was positively perceived and received by the participants. Kulier, Hadley [45] developed and evaluated the outcomes of an e-learning course for Evidence-Based Medicine training in postgraduate medical education in different languages and settings across five European countries. Students’ knowledge and attitudes were evaluated, and it was discovered that the knowledge scores significantly improved. The participants felt confident about the program paving the way for developing an international e-EBM course. Garrett and Jackson [46] developed and evaluated a wireless Personal Digital Assistant (PDA) based clinical learning tool to support and improve clinical learning, promote reflective learning in practice and help contextualize and embed clinical knowledge while in the workplace, among other objectives. The study revealed positive attitudes toward using PDA-based tools and portfolios.

On the contrary, though, Jang, Hwang [47] stated that there were no significant differences in the level of satisfaction and motivation to learn among the nursing students who participated in the web-based electrocardiography (ECG) module and those who participated using the traditional lecture method. The study discovered that the module knowledge was significantly lower in the e-learning group. However, the web-based group could interpret ECG recordings better than the group that learned traditionally. [48] described an effort to enhance medical readiness of a total program of international cooperation and conventions facilitated by the International Atomic Energy Agency. The study states that leveraging telecommunications technology as part of a training activity in radiation accident readiness can address gaps in training in this field.

#### 3.2.6. Factorial Map of the Most Cited Papers

Figure 15 provides an analytical insight into factor analysis to reveal the most cited papers. In cluster one, four papers are detected, while cluster two detects three papers. Table 5 shows the details of these papers.

### 3.3. Co-Author Analysis

We used Citation analysis to gain insight into the relative impact of the retrieved documents and Co-word analysis to find connections among concepts that co-occur in document titles, keywords, or abstracts. However, these two analyses are unable to recognize the networks of interconnections among scholars. Hence, Co-author analysis was used to analyze the authors’ collaboration and the social structure of the field. 

Author’s Appearance and Collaboration

The retrieved data (Table 6) shows 1598 authors and 1702 author appearances. There were 21 authors of single-authored documents and 1577 authors of multi-authored documents. This shows a high degree of author collaboration within this domain. Of the 315 retrieved documents, only 21 were single-authored documents with 0.197 documents per author, 5.07 authors per document, 5.40 co-authors per document, and 5.36 collaboration index.

The retrieved documents were published from 23 different disciplines, including medicine, social science, health education, engineering, nursing, and computer science, to mention just a few. This shows a high degree of interdisciplinary collaboration on the subject.

To further understand the structure of the domain, a collaboration map was plotted on VOSviewer. A co-authorship analysis was conducted using authors as the unit of analysis. A full counting method was used, and the maximum number of authors per document was capped at 25 while the minimum number of documents of an author was set to 2. Of the 1533 authors identified, only 77 met the threshold. Table 7 shows the top 10 authors collaborating in the domain in terms of the frequency of occurrences and TLS, while Figure 16 shows a network visualization of authors’ collaboration.

Some of the 77 documents in the network were not connected, so we zoomed in on the most extensive set of connected items (20). Figure 17 shows the connection between these twenty authors.

Cluster 1 (color-coded red) shows the collaboration between 12 authors, while cluster 2 (color-coded green) shows the collaboration between eight authors. 

## 4. Conclusions

The e-LMED domain is witnessing a sporadic increase in publications. This increase might be attributed to the adoption of e-learning in higher education institutions due to COVID-19. Performance evaluation is one of the most critical components in assuring the comprehensive development of e-learning in medical education. Studies evaluating performance in e-LMED used different techniques, methods, and approaches for assessing students’ performance; therefore, the ubiquity of publications has resulted in the divergence of the body of knowledge in this domain. As a result, the domain has yet to achieve any form of consensus about what factors need to be assessed when evaluating performance in e-LMED. As a starting point for this challenge, this study provided a careful analysis of the existing body of knowledge in performance evaluation in e-LMED by quantifying the literature, identifying the key terms, and analyzing the extent of research collaboration in the domain. This will help researchers understand the existing landscape of performance evaluation in e-LMED and could be used as a background to investigate and quantify the efficacy of the evaluation of e-LMED and provide a solid bedrock to develop a well-validated e-LMED performance evaluation model.

### 4.1. Interpretation and Implications of the Research Results

This study revealed that authors are actively publishing and collaborating in this domain. However, most of the top publications, collaborations, countries, institutions, and journals are produced in first-world countries; hence, more publications need to be encouraged in low-income countries. Furthermore, publication in the domain is dominated by medicine, closely followed by social science fields. These publications are clustered into pre-COVID and COVID eras. Before COVID, researchers aimed to assess performance by evaluating the feasibility, perception, and outcomes of “distance learning” since e-learning was an alternative teaching method. The COVID era introduced a boom in performance evaluation in e-LMED. This is because e-learning was implemented as Emergency Remote Learning (ERL); hence there was no proper planning, adequate need analysis, and sound program design before the migration to e-learning. Notably, performance has been a central theme across the various disciplines in medical education since COVID, and the critical terms linked most closely to performance are pre-test and post-test. This is confirmed by the factorial analysis of papers with the highest contributions and the most cited paper, revealing that most researchers analyzed performance in e-LMED by conducting pre-test and post-test of e-learning interventions. Most researchers evaluating student performance in e-LMED during the COVID era evaluated constructs such as efficacy, knowledge gain, student perception, confidence level, acceptability, feasibility, usability, and willingness to recommend e-learning. 

They have additionally evaluated performance in e-LMED without using any framework. Instead, the researchers designed their study as they deemed fit, primarily using pre-test and post-test methods and surveys. 

We recommend more publications from researchers in the computer science discipline as they currently contributed less than three percent of the retrieved documents. This will ensure that the trends in performance evaluation in e-learning in medical education are evaluated using more advanced data analysis and visualization tools to understand patterns and trends in the data better. For example, machine learning algorithms or more advanced visualization techniques could help identify more subtle patterns in the data. Also, further inter-disciplinary collaboration will lead to adopting recent data analytic approaches to performance evaluation.

The findings of this study can be leveraged to create a solid foundation upon which further studies may be conducted. We remarkably suggest further studies to analyze the Critical Success Factors (CSF) of performance evaluation of e-learning in medical education and to develop a systematic framework for the design, monitoring, and evaluation of performance in e-LMED.

### 4.2. Limitations and Future Research

The study is based on a bibliometric review of published papers. Thus, it is only as reliable as the studies that were included in the review. If there are important studies that were not included in the review, the results of the study may be incomplete or biased. The study only searched the SCOPUS database, which may not include all relevant studies despite it being so extensive. Therefore, the results of the study may not be representative of the entire field of e-learning performance evaluation in medical education.

The study used several data analysis and visualization tools, but they may not capture all aspects of the data. There may be important patterns or trends in the data that are not captured by the tools used in the study. 

## Figures and Tables

**Figure 1 healthcare-11-00232-f001:**
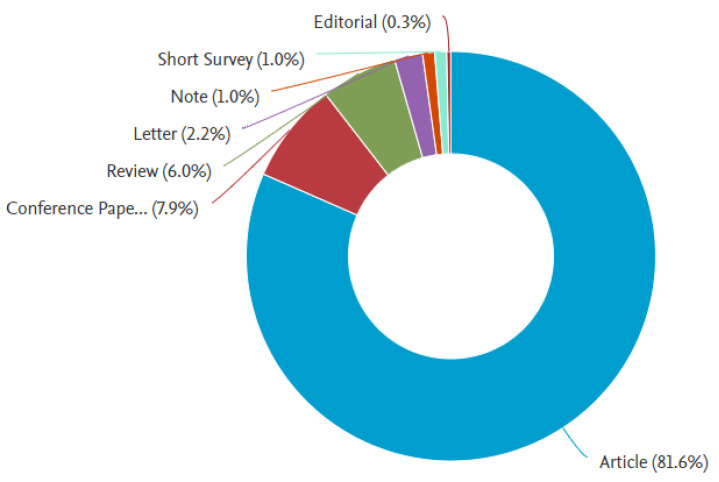
Documents by Type.

**Figure 2 healthcare-11-00232-f002:**
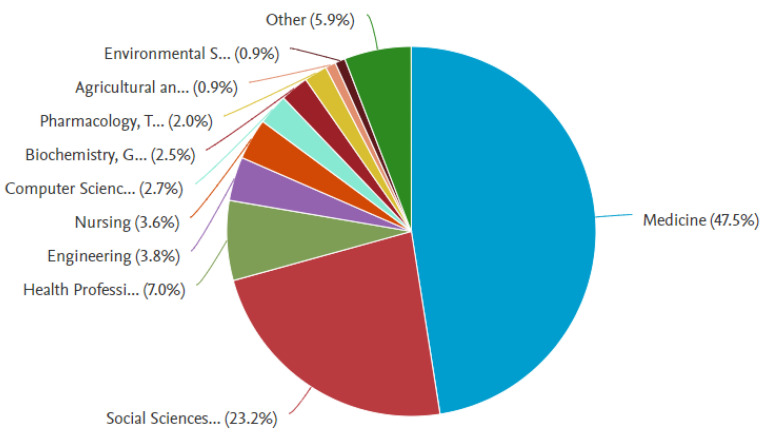
The document by Discipline.

**Figure 3 healthcare-11-00232-f003:**
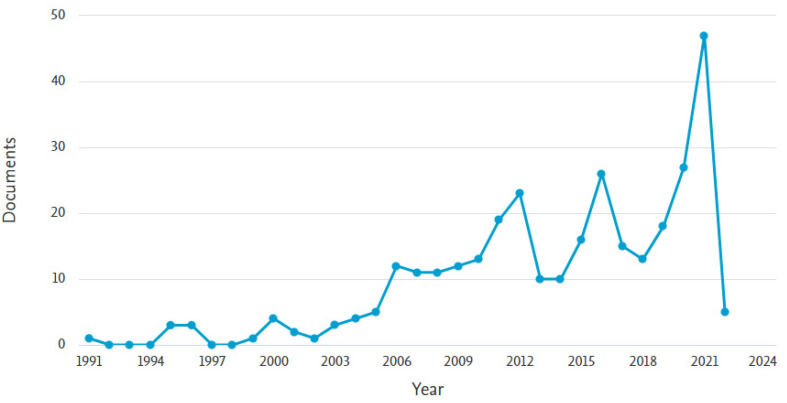
Annual publication growth of Performance evaluation of e-LMED.

**Figure 4 healthcare-11-00232-f004:**
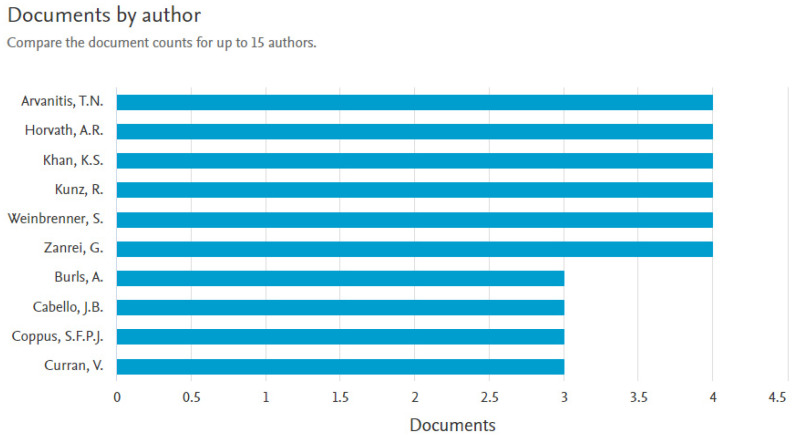
Top publishing authors in Performance evaluation of e-LMED.

**Figure 5 healthcare-11-00232-f005:**
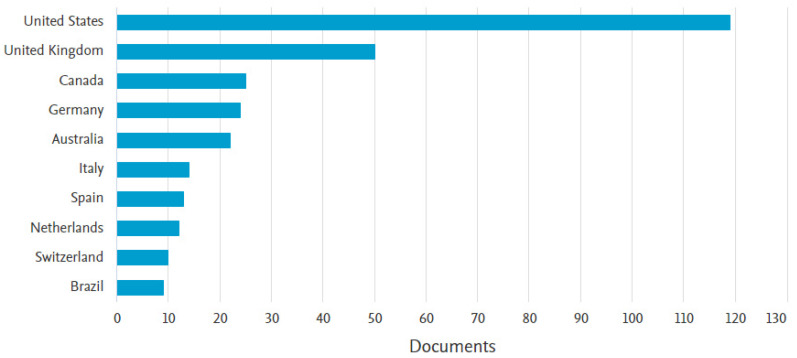
Top ten publishing countries.

**Figure 6 healthcare-11-00232-f006:**
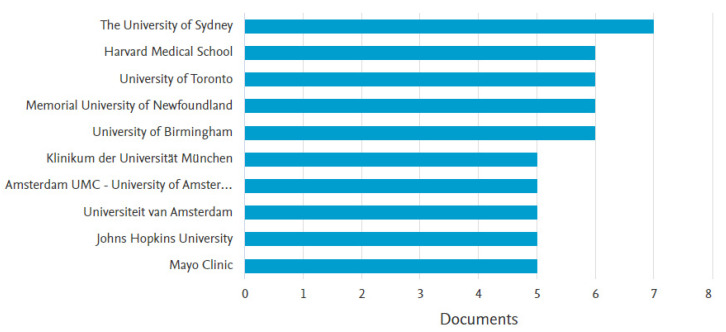
Top ten most active institutions.

**Figure 7 healthcare-11-00232-f007:**
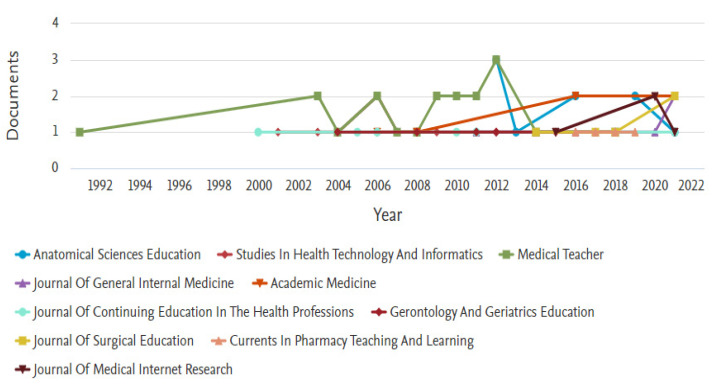
Top ten most active Journals.

**Figure 8 healthcare-11-00232-f008:**
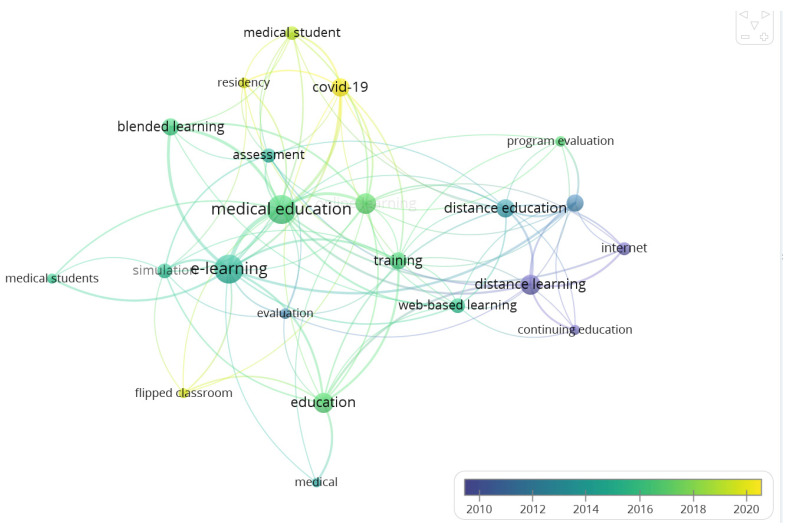
Overlay visualization map of the top 20 frequent author keywords.

**Figure 9 healthcare-11-00232-f009:**
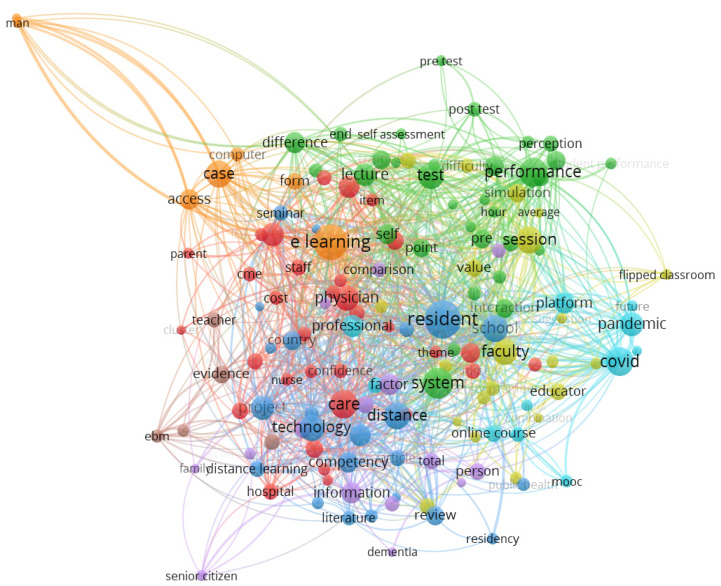
A Network Visualization of Key Terms in performance Evaluation of e-LMED.

**Figure 10 healthcare-11-00232-f010:**
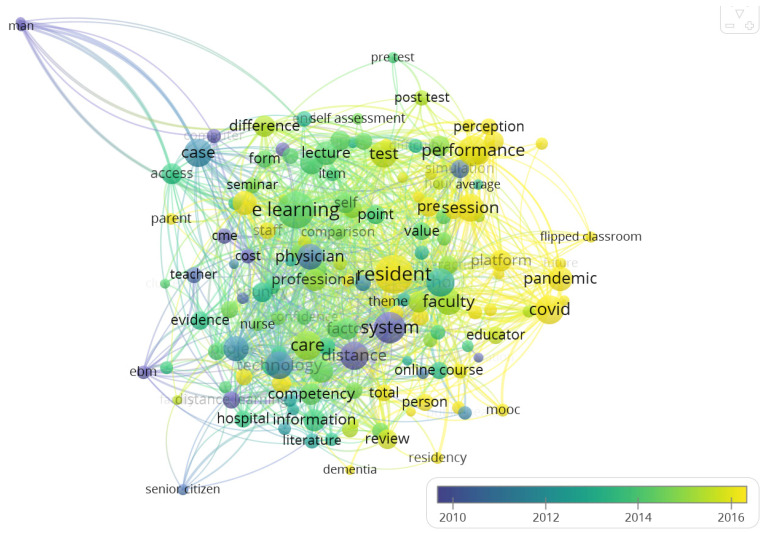
An Overlay Visualization of Key Terms in performance Evaluation of e-LMED.

**Figure 11 healthcare-11-00232-f011:**
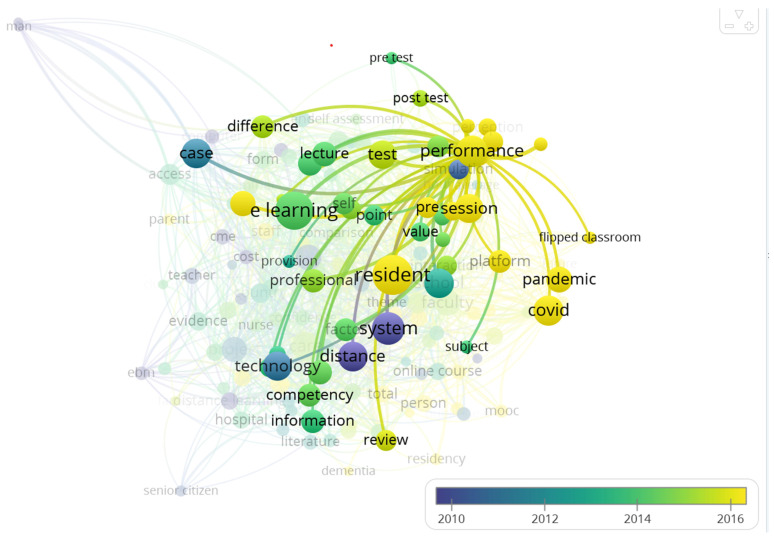
Terms co-occurring with performance.

**Figure 12 healthcare-11-00232-f012:**
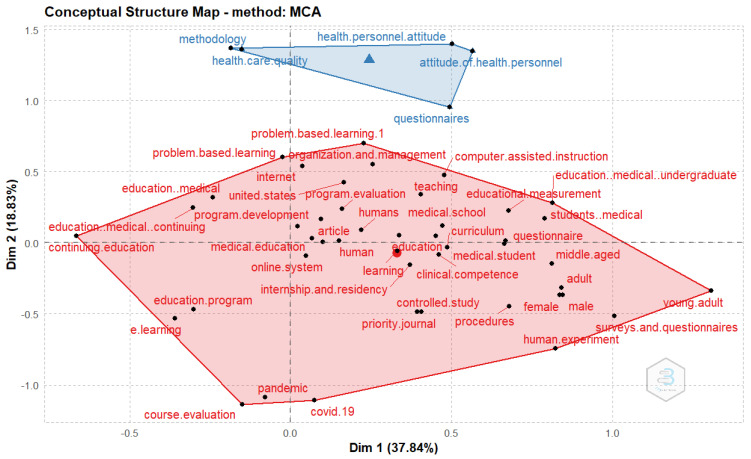
Factorial analysis using the Multiple Correspondence Analysis (MCA) method.

**Figure 13 healthcare-11-00232-f013:**
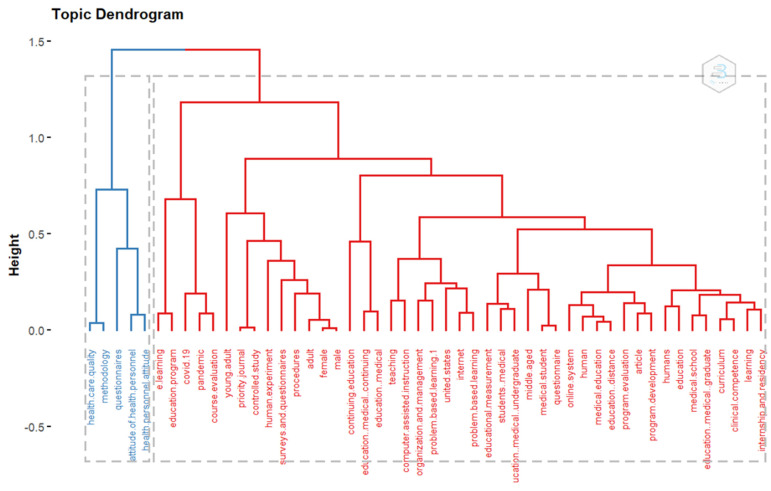
Dendrogram of hierarchical cluster analysis of keywords displaying the closeness of association between domain keywords.

**Figure 14 healthcare-11-00232-f014:**
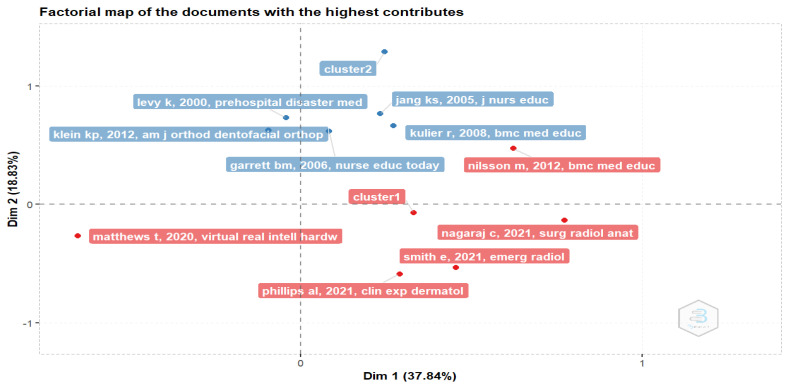
Factorial map of the document with the highest contributions.

**Figure 15 healthcare-11-00232-f015:**
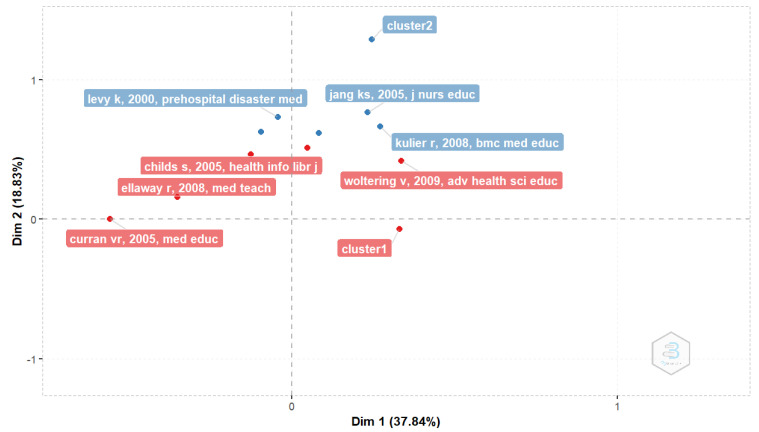
Factorial map of the most cited documents.

**Figure 16 healthcare-11-00232-f016:**
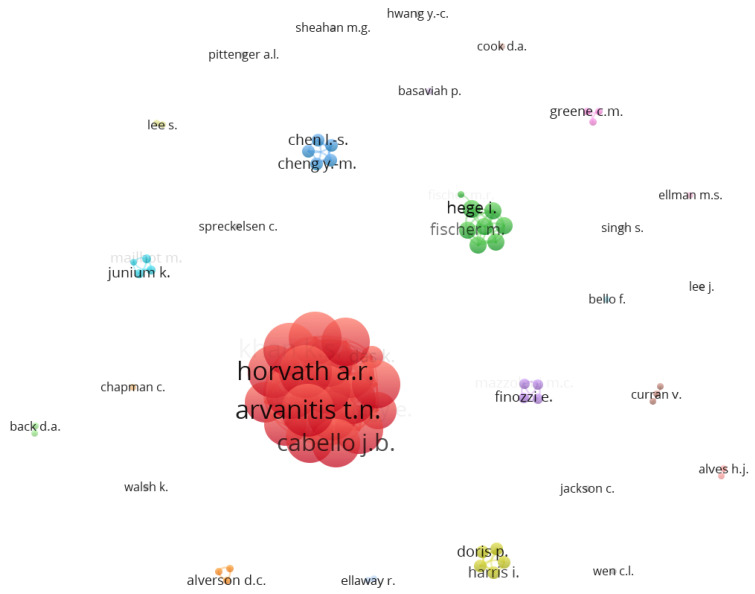
Network Visualization map of authors’ collaboration in performance evaluation in e-LMED.

**Figure 17 healthcare-11-00232-f017:**
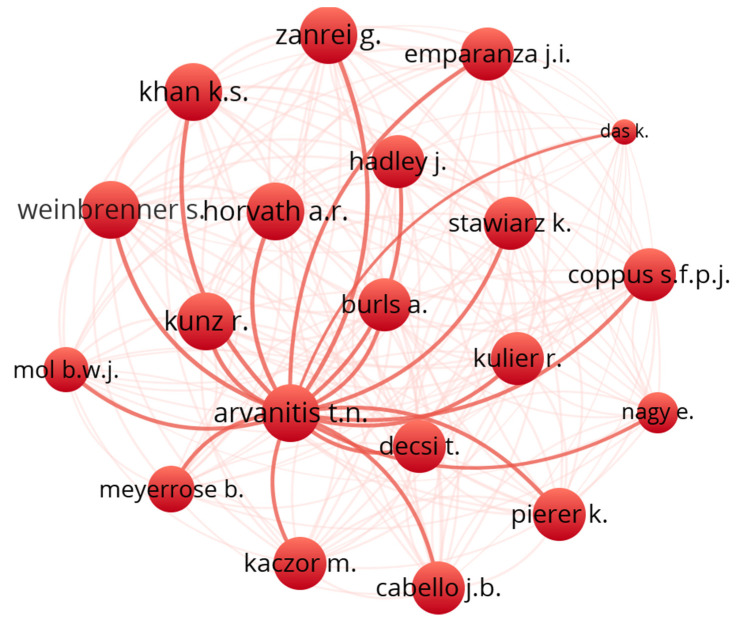
Network Visualization map of the largest set of authors’ collaboration in performance evaluation in e-LMED.

**Table 1 healthcare-11-00232-t001:** Inclusion criteria.

Criteria for Including Studies in the Review	
Population, or participants and conditions of interest	E-learning; any intervention, course, program, or module run online. The population will examine papers from all over the world
Interventions or exposures	Performance Evaluation: Any construct, factors, methods, or models used to assess e-learning interventions
Comparisons or control groups	Not applicable
Outcomes of interest	Not applicable
Setting	Medical Education; any program, module, training, or intervention in medicine.

**Table 2 healthcare-11-00232-t002:** Search Criteria.

Search Criteria		No of Documents
Population	e-learning OR “online learning” OR “online training” OR “web-based learning” OR “virtual learning” OR “digital learning” OR “digital training” OR “distance learning”	140,245
Interventions	“Performance assessment” OR “assessment of the effectiveness,” OR “assessing the performance”, OR “performance appraisal”, OR “evaluation of results”, OR “evaluation of effectiveness”, OR “evaluation of efficiency” OR “course evaluation” OR “program evaluation” OR “programme evaluation” OR “performance evaluation” OR “success factor” OR “module evaluation”	295,612
Setting	“Medical education” OR “medical training” OR “medical science” OR “medical school” OR “medical practitioner” OR “medical specialist”	395,895
Population + Setting	(e-learning OR “online learning” OR “online training” OR “web-based learning” OR “virtual learning” OR “digital learning” OR “digital training” OR “distance learning”) AND (“Medical education” OR “medical training” OR “medical science” OR “medical school” OR “medical practitioner” OR “medical specialist”)	5682
Population + Setting + Intervention	((“performance assessment” OR “assessment of the effectiveness” OR “assessing the performance” OR “performance appraisal” OR “evaluation of results” OR “evaluation of effectiveness” OR “evaluation of efficiency” OR “course evaluation" OR “program evaluation” OR “programme evaluation” OR “performance evaluation” OR “success factor OR “module evaluation”) AND (e-learning OR “online learning" OR “online training” OR “web-based learning” OR "virtual learning" OR “digital learning” OR “digital training” OR “distance learning”) AND (“medical education” OR “medical training” OR “medical science” OR “medical school” OR “medical practitioner” OR “medical specialist”))	315

**Table 3 healthcare-11-00232-t003:** The top 20 keywords with the highest TLS Value.

Rank	Term	Occurrence	TLS
1	E-learning	69	56
2	Medical Education	45	56
3	Online learning	25	28
4	Distance Learning	25	27
5	Education	26	27
6	COVID-19	16	23
7	Distance Education	20	22
8	Blended Learning	12	20
9	Continuing Medical Education	13	19
10	Training	13	19
11	Simulation	8	15
12	Web-based Learning	6	15
13	Assessment	7	13
14	Medical Student	6	12
15	Internet	7	11
16	Evaluation	6	8
17	Program Evaluation	5	8
18	Residency	5	8
19	Continuing Education	5	7
20	Flipped Classroom	7	7

**Table 4 healthcare-11-00232-t004:** The 20 most pertinent terms in performance evaluation in e-LMED.

Rank	Term	Occurrence	Relevance
1	Man	10	10.26
2	Woman	10	10.11
3	Senior Citizen	13	4.06
4	EBM	19	3.68
5	Clinical Practice	17	2.63
6	Dementia	11	2.55
7	Nurse	17	2.30
8	Family	10	2.07
9	Parent	14	2.06
10	Pre-test	13	1.98
11	MOOC	18	1.83
12	Flipped Classroom	13	1.82
13	Blended learning	19	1.81
14	Access	45	1.77
15	Student Performance	16	1.71
16	Fellow	16	1.63
17	Usability	10	1.61
18	Post Test	24	1.58
19	Computer	21	1.51
20	Case	82	1.49

**Table 5 healthcare-11-00232-t005:** The most cited document by factorial analysis.

	Paper Title	Year of Publication	Authors	No of Citation (Google Scholar)
**Cluster 1**	Blended learning positively affects students’ satisfaction and the role of the tutor in the problem-based learning process: results of a mixed-method evaluation.	January 2009	Woltering, Herrler [49]	422
	Building a virtual patient common.	June 2009	Ellaway, Poulton [50]	228
	Effective e-learning for health professionals and students—barriers and their solutions. A systematic review of the literature—findings from the HeXL project.	November 2005	Childs, Blenkinsopp [51]	579
	A review of evaluation outcomes of web-based continuing medical education	May 2005	Curran and Fleet [52]	403
**Cluster 2**	Harmonizing Evidence-based medicine teaching: a study of the outcomes of e-learning in five European countries	April 2008	Kulier, Hadley [45]	82
	Effect of Web-Based Teaching Method on Undergraduate Nursing Students’ Learning of Electrocardiography	January 2005	Jang, Hwang [47]	189
	An Internet-based Exercise as a Component of an Overall Training Program Addressing Medical Aspects of Radiation Emergency Management	June 2000	Levy, Aghababian [48]	14

**Table 6 healthcare-11-00232-t006:** Information about the retrieved document.

	Description	Results
Document Contents	Keywords Plus (ID)	1860
Authors	Author’s Keywords (DE)	700
	Authors	1598
	Author Appearances	1702
Authors Collaboration	Authors of single-authored documents	21
	Authors of multi-authored documents	1577
	Single-authored documents	21
	Documents per Author	0.197
	Authors per Document	5.07
	Co-Authors per Documents	5.4
	Collaboration Index	5.36

**Table 7 healthcare-11-00232-t007:** The top 10 collaborating Authors in performance evaluation in e-LMED.

Rank	Author	Documents	Citations	TLS
1	Arvanitis T.N.	4	151	59
2	Horvath A.R.	4	151	59
3	Khan K.S.	4	151	59
4	Kunz R.	4	151	59
5	Weinbrenner S.	4	151	59
6	Zanrei G.	4	151	59
7	Burls A.	3	127	52
8	Cabello J.B.	3	127	52
9	Coppus S.F.P.J.	3	127	52
10	Decsi T.	3	127	52

## Data Availability

Available online: https://www.scopus.com/results/results.uri?sort=plf-f&src=s&st1=%28%22performance+assessment%22+OR+%22assessment+of+the+effectiveness%22+OR+%22assessing+the+performance%22+OR+%22performance+appraisal%22+OR+%22evaluation+of+results%22+OR+%22evaluation+of+effectiveness%22+OR+%22evaluation+of+efficiency%22+OR+%22course+evaluation%22+OR+%22program+evaluation%22+OR+%22programme+evaluation%22+OR+%22performance+evaluation%22+OR+%22success+factor%22+OR+%22module+evaluation%22%29+AND+%28e-learning+OR+%22online+learning%22+OR+%22online+training%22+OR+%22web-based+learning%22+OR+%22virtual+learning%22+OR+%22digital+learning%22+OR+%22digital+training%22+OR+%22distance+learning%22%29+AND+%28%22medical+education%22+OR+%22medical+training%22+OR+%22medical+science%22+OR+%22medical+school%22+OR+%22medical+practitioner%22+OR+%22medical+specialist%22%29&sid=c464e209f025b8759a51d997c87d73ca&sot=b&sdt=b&sl=681&s=TITLE-ABS-KEY%28%28%22performance+assessment%22+OR+%22assessment+of+the+effectiveness%22+OR+%22assessing+the+performance%22+OR+%22performance+appraisal%22+OR+%22evaluation+of+results%22+OR+%22evaluation+of+effectiveness%22+OR+%22evaluation+of+efficiency%22+OR+%22course+evaluation%22+OR+%22program+evaluation%22+OR+%22programme+evaluation%22+OR+%22performance+evaluation%22+OR+%22success+factor%22+OR+%22module+evaluation%22%29+AND+%28e-learning+OR+%22online+learning%22+OR+%22online+training%22+OR+%22web-based+learning%22+OR+%22virtual+learning%22+OR+%22digital+learning%22+OR+%22digital+training%22+OR+%22distance+learning%22%29+AND+%28%22medical+education%22+OR+%22medical+training%22+OR+%22medical+science%22+OR+%22medical+school%22+OR+%22medical+practitioner%22+OR+%22medical+specialist%22%29%29&origin=savedSearchNewOnly&txGid=a14be460b8740f7a387c1d9c17022076 (accessed on 16 May 2022).

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
