# Peer review of "E-Learning Performance Evaluation in Medical Education—A Bibliometric and Visualization Analysis"

_healthcare, 2023, doi:10.3390/healthcare11020232_

Round 1

Reviewer 1 Report

This paper is bibliometric and visualization analysis. It is about the use of e-learning in medical education and the evaluation of its performance. It aims to analyze the scientific productivity and the impact of the published factors on the performance evaluation of e-learning in medical education. For this purpose, the authors searched the SCOPUS and found 315 studies. They then used various tools, such as R Bibliometrix, Biblioshiny, and VOSviewer, to analyze the data and visualize the results. The results of the study indicate that there has been a sporadic increase in publications on the topic of performance assessment in e-learning in medical education in 2021. The main authors, countries, institutions, and sources in this field are also identified. The study also identifies the conceptual structure of the field, mentioning topics such as healthcare quality, healthcare worker attitudes, e-learning, Covid-19, and course evaluation. The authors hope that this study will help researchers understand the existing landscape of performance evaluation of e-learning in medical education and identify any knowledge gaps in the field.

Dear authors, I appreciate the opportunity to review this paper.

I see some formal improvements as necessary and also some limitations of the study should be considered:

There are several potential limitations to this paper which shall be discussed in context in the discussion chapter:

1.       The study is based on a literature review of published papers. Thus, it is only as reliable as the studies that were included in the review. If there are important studies that were not included in the review, the results of the study may be incomplete or biased.

2.       The study only searched the SCOPUS database, which may not include all relevant studies despite it is so extensive. Therefore, the results of the study may not be representative of the entire field of e-learning assessment in medical education.

3.       The study used several data analysis and visualization tools, but they may not capture all aspects of the data. There may be important patterns or trends in the data that are not captured by the tools used in the study. Discuss other possibly omitted trends / patterns.

4.       The study only addresses performance evaluation of e-learning in medical education and does not consider other factors that may impact the effectiveness of e-learning in this field, such as the design of the e-learning course or the technical infrastructure that supports it.

There are several suggestions for improvement:

1.       The study could include more advanced tools for data analysis and visualization to better understand patterns and trends in the data. For example, using machine learning algorithms or more advanced visualization techniques could help identify more subtle patterns in the data. Possibly discuss this in the discussion.

2.       The paper can consider a broader range of factors that may impact the effectiveness of e-learning in medical education. For example, the design of the e-learning course, the technical infrastructure that supports it, or the characteristics of the learners could have an impact on the performance of e-learning in this field.

3.       The study could also provide a more detailed analysis of the biases in the published literature, in order to better understand the potential impact of these biases on the findings of the study.

4.       The study could be hypothetically extended to include a more in-depth analysis of the conceptual structure of the field, in order to better understand the key themes and trends in e-learning performance evaluation in medical education. This could involve the use of techniques such as content analysis or thematic analysis to identify key themes and patterns in the data. Generally, a more comprehensive and detailed analysis of the data would help to improve the reliability and usefulness of the findings of the study.

5.       The WEB 2.0 induced a paradigm shift in the e-learning 10 years ago and enabled e-learning scenarios that were precursors to current e-learning as a powerful and timely pedagogical tool. Please reference source regarding the WEB 2.0 induced paradigm shift in the e-learning and the role of crowdsourcing in dental education.

Figures are not properly numbered. Figure 8 is followed by 8a (the top 22 frequent author keywords) must be edited / enlarged so the text – the words are readable.

Figure 8c: shall be a table, not figure, please use the correct format prescribed in the Journal template.

The size of the text on Figure 11 shall be enlarged so it is readable. Text on the Figure 14 and 15 is unreadable.

Your paper is missing compulsory information about Author Contributions and Funding.

Please rewrite the chapter Conclusions to be briefer and clearer. In other aspects the paper is suitable for publication.

Author Response

Good day,

Thank you for your time and detailed review of our work. We value your input. I am sharing with you our responses to each of your queries.

We hope you find it worthwhile. 

Kind regards,

Deborah Oluwadele

---

Dear Reviewer#1

I see some formal improvements as necessary and also some limitations of the study should be considered:

There are several potential limitations to this paper which shall be discussed in context in the discussion chapter:

  1. The study is based on a literature review of published papers. Thus, it is only as reliable as the studies that were included in the review. If there are important studies that were not included in the review, the results of the study may be incomplete or biased.
  2. The study only searched the SCOPUS database, which may not include all relevant studies despite it is so extensive. Therefore, the results of the study may not be representative of the entire field of e-learning assessment in medical education.
  3. The study used several data analysis and visualization tools, but they may not capture all aspects of the data. There may be important patterns or trends in the data that are not captured by the tools used in the study. Discuss other possibly omitted trends / patterns.
  4. The study only addresses performance evaluation of e-learning in medical education and does not consider other factors that may impact the effectiveness of e-learning in this field, such as the design of the e-learning course or the technical infrastructure that supports it.

I have added a section for limitation of the study and included the points you have raised as limitations of the study.

There are several suggestions for improvement:

  1. The study could include more advanced tools for data analysis and visualization to better understand patterns and trends in the data. For example, using machine learning algorithms or more advanced visualization techniques could help identify more subtle patterns in the data. Possibly discuss this in the discussion.

The use of machine learning algorithm would definitely be great for this study. We have included this as recommendation for future works as this is beyond the scope of our study 774-778.

  1. The paper can consider a broader range of factors that may impact the effectiveness of e-learning in medical education. For example, the design of the e-learning course, the technical infrastructure that supports it, or the characteristics of the learners could have an impact on the performance of e-learning in this field.

We are currently conducting another study using the systematic review of literature to answer the questions raised here. For the purpose of this study, this is beyond our scope. Our aim is to understand the skeletal structure of the domain; hence we use bibliometric analysis and not systematic literature review for this current study.

  1. The study could also provide a more detailed analysis of the biases in the published literature, in order to better understand the potential impact of these biases on the findings of the study.

We would consider this in our next study.

  1. The study could be hypothetically extended to include a more in-depth analysis of the conceptual structure of the field, in order to better understand the key themes and trends in e-learning performance evaluation in medical education. This could involve the use of techniques such as content analysis or thematic analysis to identify key themes and patterns in the data. Generally, a more comprehensive and detailed analysis of the data would help to improve the reliability and usefulness of the findings of the study.

Please look at line 470-625, we used Multiple Correspondence Analysis to understand the conceptual structure of the domain. The topic dendrogram 495-523 was used to identify keywords and group the keywords into two clusters using the content of the retrieved document. Also we used co-word analysis 347-469 to analyse the key themes and pattern in the data of the domain.

  1. The WEB 2.0 induced a paradigm shift in the e-learning 10 years ago and enabled e-learning scenarios that were precursors to current e-learning as a powerful and timely pedagogical tool. Please reference source regarding the WEB 2.0 induced paradigm shift in the e-learning and the role of crowdsourcing in dental education.

Included in line 109-114

Figures are not properly numbered. Figure 8 is followed by 8a (the top 22 frequent author keywords) must be edited / enlarged so the text – the words are readable.

We have done this. Please let us know if the readability of the figures has now improved.

Figure 8c: shall be a table, not figure, please use the correct format prescribed in the Journal template.

Done 437

The size of the text on Figure 11 shall be enlarged so it is readable. Text on the Figure 14 and 15 is unreadable.

We have done this. Please let us know if the readability of the figures has now improved.

Your paper is missing compulsory information about Author Contributions and Funding.

This has been sorted out in line 779-782.

Please rewrite the chapter Conclusions to be briefer and clearer.

Done

 In other aspects the paper is suitable for publication.

We really appreciate you.

Reviewer 2 Report

Attached 

Author Response

Good day,

Thank you for your time and detailed review of our work. We value your input. I am sharing with you our responses to each of your queries.

We hope you find it worthwhile. 

Kind regards,

Deborah Oluwadele

---

Dear Reviewer#2

The manuscript “E-Learning Performance Evaluation in Medical Education - A Bibliometric and

Visualization Analysis” performed bibliometric analysis for medical education.

  1. Why did the authors select the Scopus database for proposed manuscript? In medical

studies, what are the databases available except MEDLINE?

Scopus was used because it is 100% inclusive of MEDLINE and has a more significant number of indexed journals than other databases. Also, SCOPUS has many functions that can be leveraged to facilitate citation analysis, counting research collaboration, and ex-porting data to Microsoft Excel for further tabulation and mapping. 86-89

  1. Why did the authors not consider the PubMed and Web of Science databases for the study?

WoS database gives many documents as compared to your results.

Scopus was used because it is 100% inclusive of MEDLINE and has a more significant number of indexed journals than other databases. Also, SCOPUS has many functions that can be leveraged to facilitate citation analysis, counting research collaboration, and ex-porting data to Microsoft Excel for further tabulation and mapping.

  1. For the search query, what are the others settings? Means years have been considered in

this proposed work.

315 studies published between 1991 and 2022 were retrieved 19. This has been added

  1. Why does the Biblioshiny tool used in this study? The same functions can be done using

VOSViewer.

Biblioshiny helped us do analysis such as factorial analysis using multiple correspondence analysis and topic dendogram. This feature is not available on VosViewer. We used both applicatioons to make up for the inadequacies of each other.

  1. Is it NOT too few documents retrieved from the search query?

The Guidelines for using bibliometrics at the Swedish Research Council stipulated that for pure bibliometric comparisons, the publication data should exceed at least 50 articles, while results which are to be used by subject experts along with other information can be used if the sample size is more than 20 articles (reference).

  1. Figures can be centered for better readable. Done
  2. “However, publication in this domain continued to grow yearly after that and 248 witnessed

sporadic growth in 2021 (n=47, _ %);” - why is it a blank percentage? Resolved

  1. “Arvanitis, T. N. a professor of Digital Health Innovation and 258 Director of the Institute of

Digital Healthcare” - check his google scholar page to validate your output.

I have crosschecked and confirmed his publication relative to our inclusion criteria and search term.

  1. “Figure 3 illustrates the top ten most actively publishing countries in performance 265

evaluation of e-LMED.” - check this, it should be figure 5, right?

Thank you loads. I have corrected the error.

  1. What about links and TLS for your result analysis using VOSViewer?

We have included the link in the data availability section.

Reviewer 3 Report

Dear Authors,

I have read with attention your submitted paper titled 'E-Learning Performance Evaluation in Medical Education - A 2 Bibliometric and Visualization Analysis'.

The presentation of the data is attractive.

We have spotted only few typos.

The list of the analized works, as well as the dataset, are not attached and should be provided in order to verify the consistency of the results.

The topic is interesting and the findings could be a solid basis 

Author Response

Good day,

Thank you for your time and detailed review of our work. We value your input.

Based on your recommendation, we have reworked our research design using the PICOS model to inform our inclusion criteria and search terms.  We have also carried out grammar checks for errors that we identified.

We hope you find our corrections worthwhile. 

Kind regards,

Deborah Oluwadele

Round 2

Reviewer 1 Report

As requested, the size of the text on Figures was not enlarged and still remains unreadable. For example the text on the Figure 14 and 15 seems unchanged. Figure 15 is not shown in the PDF.

Neither was edited the Figure 11 text size and other figures.

You have also not red carefully the remark regarding Figure 8c. you have remade it to a table however you did not used the correct format as is prescribed in the template - instructions for authors. It is unknown why you have not formatted the table 3 properly so again, please use the correct format prescribed in the Journal template for table 3. 

Also regarding the Web 2.0 induced a paradigm shift in the e-learning from 10 years ago, you did reference a new work which is valuable, however this paper describes only two Australian case studies, with an ex-poste evaluation of the use of Web 2.0 tools. As you are introducing it in the Introduction chapter in the context of a change delivered 10 years ago I recommend to  reference work from that time for example: https://pubmed.ncbi.nlm.nih.gov/20437831/
http://bmj.fmed.uniba.sk/2010/11103-14.pdf  - 

The WEB 2.0 induced paradigm shift in the e-learning and the role of crowdsourcing in dental education

In your text "The advent of Web 2.0 induced a paradigm shift in the e-learning ten years ago and enabled e-learning scenarios that were precursors to current e-learning...
Included in line 109-114"

Author Response

Good day,

We thank you for your review. All points are fully noted and reworked. We have scaled the texts of the images for more visibility. Hence the results are better presented. We have also ensured that all tables use the prescribed format.

We also used the reference you provided for web 2.0. We thank you for providing the reference.

We hope you find everything in order.

Kind regards,

Deborah Oluwadele

Reviewer 2 Report

Need to show link strength in bibliometric analysis. Other points are okay. 

See this papers

1. https://journal.umy.ac.id/index.php/jrc/article/view/15453

2. https://www.mdpi.com/2571-5577/4/4/86

Author Response

Dear reviewer,

Thank you for your time and effort. We have included TLS (lines 377,384, and 678). We found the links to the papers you provided very helpful.

We hope you find everything in order.

Kind regards,

Deborah Oluwadele

Reviewer 3 Report

Dear Authors,

we appreciated your commitment for improving the quality of your work.

We would like to give you further suggestions.

Lines 21-24: Top publishing authors as well as top publishing Institutions are not necessary in the Abstract. Please delete.

Please check also the presence of the authors' names of the cited papers and insert them wherever missing (e.g. line 140 etc.) 

The aim of the article is just to quantify the amount of research on the topic,  identify the key terms and analyze the extent of research collaboration in that domain.
In this frame, the results are well presented but they do not provide any relevant knowledge increase. If used as a background in order to investigate and quantify the efficacy of the evaluation of e-Learning, it could be much more interesting. May you provide additional implementation about this topic in the result section and underline it in the discussion?

Author Response

Dear reviewer,

We thank you for your valuable contributions and reviews. We have effected all your suggestions. We reworded the abstract and conclusion. We also rephrased the aim of the study. We also checked that all references are on point.

We provided additional implementation about the topic, as you suggested. This is reflected in our reworded abstract. We emphasized that the study could be used as a background to investigate and quantify the efficacy of the evaluation of e-Learning. We underline this in the conclusion section. Please refer to the initial cover letter attached below to see how this study will fit into our overall objective to design a performance evaluation framework in e-LMED. We have four more articles to publish in this regard.

We hope you find everything in order.

Kind regards,

Deborah Oluwadele
